

# The charcoal morphology

# of different vegetation types in the wildfire

Shurong Zhang [1,2], Hui Shen [1,2], Xinying Zhou [1,2], Ming Ji [3], Xiaoqiang Li [1,2]

5   [1] Key Laboratory of Vertebrate Evolution and Human Origins, Institute of Vertebrate
Paleontology and Paleoanthropology, Chinese Academy of Sciences, Beijing, 100044,
China
[2] University of Chinese Academy of Sciences, Beijing, 100049, China
[3] School of Chemistry, Biology and Environment, Yuxi Normal University, Yuxi,
10   Yunnan, 653100, China

Correspondence to: Hui Shen (shenhui@ivpp.ac.cn)




**Abstract.** Wildfires are a crucial element in the earth's ecosystem, playing a significant role in maintaining ecological balance. Charcoal, a key product of fires, can be examined for its morphological traits to distinguish between various fuel sources. This

analysis helps in understanding the relationship between regional vegetation environments and fire dynamics. However, current research on charcoal morphology heavily relies on simulation experiments, lacking real-world data from natural wildfires. This study delves into the morphological characteristics of charcoal formed under wildfire conditions by investigating samples from four distinct vegetation types. By

sidestepping the uncertainties of simulation experiments, the research uncovers intriguing patterns. The findings indicate a gradual decrease in charcoal size across different ecosystems, such as evergreen coniferous forests, warm-humid herbs, evergreen broad-leaf forests, and grasslands. The length-to-width (L/W) ratio of charcoal alone is insufficient for classifying fuel types accurately. To address this issue,

a decision tree model was crafted to effectively categorize various charcoal types, achieving an impressive accuracy rate of 72.44%. This paper presents a thorough examination of charcoal morphology within diverse vegetation landscapes, offering valuable insights for fuel type identification based on charcoal characteristics. Ultimately, this research contributes to enhancing our knowledge of the intricate

interactions between natural wildfires and vegetation dynamics.





# 1 Introduction

Fire plays a crucial role in natural ecosystems, with significant impacts on the environment (Glasspool et al.,2004; Bowman et al.,2009; Scott et al.,2013,2018). On the one hand, it recognized for its contribution to carbon emissions and its role in global warming through the release of $CO_2$ and affecting terrestrial biodiversity (Marlon et al.,2009; Bowman et al.,2011; Karp et al.,2021). Conversely, wildfires have a direct

influence on the patterns and evolutionary processes within global terrestrial ecosystems by modifying regional vegetation composition and structure, thereby participating in vegetation evolution (Kaars et al.,2000; Lundeen et al.,2016; Pyne,2019). Increasing number of studies indicate that the occurrence of wildfires is influenced by a complex interplay of climate conditions, vegetation characteristics, and

human activities (Aldersley et al.,2011; Dussol et al.,2021; Xu et al.,2021). Hence, a comprehensive understanding of the evolutionary patterns of wildfires and the intricate relationships between wildfires, climate, and vegetation is indispensable (Power et al.,2008; Keeley et al.,2005; Halofsky et al.,2020). Such knowledge can significantly enhance predictive capabilities and inform effective management strategies for

controlling fire trends within the context of impending global warming scenarios. Academic discourse underscores the importance of exploring these interconnected dynamics to facilitate improved fire management practices and adaptive measures in response to shifting environmental conditions (Brown et al.,2009; Scott et al.,2010; Lasslop et al.,2020).

Charcoal, as a direct product of fire, possesses unique properties such as high-temperature resistance, ease of preservation, high yield and widespread distribution.

It serves as a crucial indicator for investigating the evolutionary history of paleo-fires during different geological periods and shedding light on the frequency and intensity of wildfires (Umbanhowar and McGrath,1998; Masiello,2004; Hawthorne et

al.,2018; Feurdean,2021). Researchers increasingly focused on wildfires studying across various geological eras globally by analyzing charcoal from marine, lake and loess sediments (Hoetzel et al.,2013; Mustaphi and Pisaric,2014; Tan et al.,2015).



Charcoal, resulting from the incomplete combustion of plants, offers valuable insights into original plants anatomy through its morphology, surface texture and pore features

(Crawford et al.,2014; Dussol et al.,2021). Being chemically and biologically inert, the morphological features of charcoal be used to discern the source plants about the climate and vegetation environment during the specific time period. Furthermore, these features can also provide clues about the climate and vegetation environment during the specific time (Zhang and Lv,2006; Mindrescu et al.,2023; Feurdean et al.,2023). For

instance, Enache and Cumming (2006) have categorized charcoal into seven types based on parameters such as morphology, aspect ratio and porosity, and they also analyzed the relationship between charcoal morphology, they have also explored the relationship between charcoal morphology, fire types and precipitation. In recent years, an increasing number of researches have delved into multidimensional studies on

charcoal morphology (Thevenon and Anselmentti,2007; Hu et al.,2020; Dussol et al.,2021). Recent studies on charcoal morphology have highlighted the significance of length-width (L/W) ratio of charcoal as a valuable proxy for distinguishing between various fuel types. Umbanhowar and McGrath (1998) observed distinct characteristics between herbaceous and woody charcoal, noting that herbaceous charcoal tends to be

elongated with length and area are 563 μm and 5630 μm$^2$, while woody charcoal typically appears square or round, measuring 380 μm and 64946 μm$^2$ in length and area. Moreover, there is discernible difference in the L/W ratio, with herbaceous averaging at 3.62 and woody charcoal at 1.91. Zhang and Lv (2006) performed simulated experiments on 23 common plant species in northern China, revealing notable

differences in morphology and L/W ratio of different charcoal, despite their sizes are relatively similar. The average length and L/W ratio of the woody charcoal were reported to be 103.6±9.4 μm and 3.1±0.2, respectively, while the herbaceous charcoal exhibited an average length and L/W ratio of 109.0±11.4 μm and 10.2±1.3. Similarly, Crawfords and Belcher (2014) undertook similar experiments on European plants,

where they explored the influence of river transport on charcoal morphology through simulations involving water and gravel. Their findings were consistent with those



reported by Zhang and Lv (2006), indicating distinct characteristics in the morphology and L/W ratio of charcoal derived from different plant species. These studies highlight the nuanced variations in charcoal morphology based on plant source and

environmental factors, underscoring the importance of understanding these complexities in interpreting charcoal samples from different regions and ecosystems.

While previous studies have relied on laboratory simulation experiments, which cannot fully replicate the burning conditions of actual wildfires, there exists a notable gap in systematic research on the charcoal morphology arising from various plants

under natural wildfire scenarios. The distribution characteristics of key charcoal parameters, including length, width and area remain ambiguous, and potential for these parameters to function as proxies for distinguishing between different fuel types is a subject of ongoing debate. Despite observed differences in L/W ratio of charcoal from distinct plants, specific thresholds for classification have yet to be established.

Therefore, further extensive research is warranted in this area. To address these issues, this paper undertakes the collection of charcoal samples resulting from diverse types of vegetation in modern mountain fires. All the charcoal were extracted using a standardized pollen methodology, followed by photographic documentation and statistical analysis to unveil the morphological features of charcoal from various

vegetation types in natural wildfire scenarios. Subsequently, the parameters of charcoal, such as length, width, and area, were meticulously measured to enable a comprehensive and systematic characterization of charcoal attributes. Moreover, a decision tree classification model was utilized to effectively differentiate between different types of charcoal, providing valuable insights for analyzing fuel sources within natural wildfire

contexts. This approach offers essential reference data for enhancing our understanding of charcoal morphology and its potential applications in the identification and interpretation of fuel types in wildfire occurrences.

## 2 Materials and methods

In this paper, a total of 10 charcoal samples were collected from Sichuan, Yunnan,



Guizhou and Inner Mongolia Autonomous Region (Fig. 1), which were classified into three groups according to vegetation types, including 4 evergreen coniferous forests, 1 evergreen broad-leaf forest, and 1 grassland. In addition, 4 warm-humid herbs were also included (Table 1).

All the charcoal samples were extracted using a standardized pollen methodology.

The specific procedure was as follows: 1) 10 samples of 100 g were taken and placed them in the breakers, with enough hydrochloric acid (HCl) added to remove carbonas; 2) water was added to the breakers after complete reaction, then placed the breakers on a hot plate for heating. After a period of reaction time, the beakers were removed from the hot plate and two lycopodium spore tablets (10315 grains/tablet) were added,

followed by water, allowing the mixture to stand for 12 hours; 3) The supernatant was poured off, and the samples were transferred to the centrifuge tubes for drying; 4) The samples were stirred and centrifuged after adding zinc iodide solution with a specific gravity of 1.98, and the supernatant was collected in breakers; 5) A specific amount of water and glacial acetic acid was added to the beakers, and the mixture was allowed to

stand for 12 hours; 6) The samples were collected into centrifuge tubes, heated in a water bath with a mixture of sulfuric acid and acetic anhydride, then washed with water until neutral, then dried them; 7) The prepared samples were followed by photographic documentation and statistical analysis. Each sample was counted and photographed under the light microscope at 400X magnification (Zeiss Axiolmager.A2).

The photographs of charcoal were imported into Image J for identification, measurement and analysis. The measured parameters included length, width and area. The process was as follows: 1) Convert the original color images to 8-bit images using the command Image—Type—8-bit; 2) Set the scale; 3) Adjust the threshold using the command Image—Adjust—Threshold to align with the edges of the samples to be

measured; 4) Obtain the area of the measured samples using the command Analyze—Measure; 5) Select the charcoal to be measured and use the command Edit—Selection—Fit Rectangle to determine the length and width of the samples. At least 250 particles were counted for each sample. Additionally, charcoal was selected under the



microscope and transferred to the stage using appropriate tools. The samples were then
subjected to gold sputtering and observed under the scanning electron microscope
(Zeiss EVO25) at various magnifications to capture morphological features and take
photographs.

## 3 Results

### 3.1 Charcoal morphology

The evergreen coniferous forests are divided into four groups, and the dominant
species are *Pinus yunnanensis, Cryptomeria japonica, Cryptomeria japonica* and *Pinus
densata.* The undergrowth is accompanied by some small shrubs, and the surface debris
is mainly covered with fallen leaves and a few small dry branches. Charcoal deposited
on the ground surface after wildfire is primarily composed of remnants plant tissues
and undergrowth debris, involving leaves, bark and branches. Only the bark of the
trunks underwent combustion, while the internal secondary phloem and xylem
remained unaffected. Observations indicate that charcoal resulting from the
experimental treatment is polygynous, flaky with flat edges and smooth surface (Fig.
2).

The evergreen broad-leaf forests are dominated by *Quercus glauca*, with almost
no accompanying shrubs in the undergrowth. The surface debris primarily consists of
fallen leaves and branches. Charcoal is mainly originating in the plant tissues burning
from *Quercus glauca*, including leaves and branches. Only the bark of the trunks
underwent combustion, while the internal secondary phloem remained unaffected. The
charcoal morphology after experimental treatment resembles that arising from the
burning of evergreen coniferous forests, which exhibits polygynous shape and appears
flaky or strip-shaped with clear and smooth edges. But the size is greatly smaller than
that of charcoal derived from coniferous forests (Fig. 3).

The grasslands are primarily dominated by the *Stipa capillata*, which is devoid of
large woody plants and consisting mainly of low-growing tufted plants. Following a





natural wildfire, the charcoal remaining on the surface primarily originates from *Stipa capillata*. The leaves and stems were combusted completely, just leaving the charred basal stems. And the level of char coalification process was heavily higher than that observed in evergreen coniferous and broad-leaf forests. The charcoal characteristics

are predominantly elongated, smaller in size, and presenting punctate or granular with irregular edges (Fig. 4).

The warm and humid herbs from four groups, dominated by *Miscanthus sinensis*, *Neyraudia reynaudiana*, *Arundinella setosa* and *Cyperus cuspidatus*, respectively. This vegetation type is characterized by relatively tall plants with robust stems. When

exposed to natural wildfires, it underwent complete combustion, only leaving charred, sturdy stems behind, while remaining parts are entirely consumed. This process results in a high degree of char-coalification. The charcoal features are primarily rectangular or elongated, with jagged and irregular edges. Some larger elongated charcoal occurs as bundled aggregates (Fig. 5).

**3.2 The size of charcoal**

**3.2.1 The length of charcoal**

Measuring the parameters of charcoal after photographing under the microscope, including length, width, area and L/W ratio (Table 2).

In the evergreen coniferous forests, the charcoal of *Cryptomeria japonica* appears

the greatest length values (mean = 9.29±6.67; range =2.18-43.22) (at a 95% confidence level, the same below) (Table 2) (Fig. 6). The charcoal of *Pinus densata* exhibits the lowest length values (mean = 6.30±3.56; range =2.12-29.02). The average lengths of *Pinus yunnanensis* (mean = 7.66±4.43; range =2.19-31.62) and *Pinus massoniana* (mean = 7.73±5.05; range =2.21-36.67) are relatively close. The length distributions for

all four groups are predominantly concentrated in the 0-10 μm, accounting for 78%, 68%, 79%, and 88% in *Pinus yunnanensis, Cryptomeria japonica, Pinus massoniana* and *Pinus densata.*, respectively. The secondary range of 10-20 μm comprises 20%,



26%, 17%, and 12%. The proportions of all remaining distribution intervals constitute less than 10%. Statistical analysis reveals no great difference in the length of charcoal between *Pinus yunnanensis* and *Pinus massoniana,* whereas the discrepancy exists among the other species (ANOVA; P<0.05).

The charcoal length of the evergreen *Quercus glauca* forests (mean = 5.84±3.24; range =1.85-21.52) is mainly concentrated in the 0-10 μm (90%) and 10-20 μm (9%).

The vast majority of charcoal length from *Stipa capillata* grasslands (mean = 3.32 ±1.46; range =1.26-10.83) is concentrated in the 0-10 μm.

The charcoal length data pointed distinct patterns among the warm-humid herbs. Neyraudia reynaudiana (mean = 7.01±4.16; range =2.34-25.02) and Arundinella setosa (mean = 7.01±4.59; range =2.07-31.66) show the maximum values, while Miscanthus sinensis (mean = 4.57±2.30; range =1.85-16.40) exhibits the minimum value. Cyperus cuspidatus (mean = 6.46±3.74; range =1.66-27.08) displays intermediate value. The length ranges for all four groups are focus on the 0-10 μm, with percentage of 96%, 88%, 84%, and 83%, for Miscanthus sinensis, Cyperus cuspidatus, Neyraudia reynaudiana and Arundinella setosa. And the 10-20 μm accounted for 4%, 11%, 14%, and 15%, while the other distribution interval are all less than 5%. Only the Miscanthus sinensis differs significantly from other species (ANOVA; P<0.01).

### 3.2.2 The width of charcoal

The measurement results demonstrate that the average width of evergreen coniferous forests is the maximum, followed by evergreen *Quercus glauca* forests, warm-humid herbs, and *Stipa capillata* grasslands in descending order. Among the evergreen coniferous forests, *Cryptomeria japonica* (mean = 5.50±3.72; range =0.93-27.89) displays the widest charcoal (Table 2), with the broadest distribution (Fig. 7). Meanwhile, *Pinus densata* (mean = 3.96±2.24; range =1.10-16.45) produces the narrowest charcoal. The average width of *Pinus yunnanensis* (mean = 4.11±2.35; range =1.16-23.07) and *Pinus massoniana* (mean = 4.59±2.64; range =1.32-16.11) fall in between. Only *Cryptomeria japonica* shows observably discrepancy in charcoal width



compared to the other vegetation types (ANOVA; P<0.01). In evergreen *Quercus glauca* forests, the average charcoal width is 3.59±1.75 μm, with a distribution range of 1.64 to 11.50 μm. In *Stipa capillata* grasslands, the average charcoal width is 1.98 ± 0.76 μm, ranging from 0.71 to 4.77 μm. Among the warm-humid herbs, *Miscanthus sinensis* (mean = 2.99±1.24; range =1.64-11.50) has the smallest average charcoal width, while *Neyraudia reynaudiana* (mean = 4.00±2.54; range =1.00-18.05) has the largest average width. The average widths of *Arundinella setosa* (mean = 3.78±2.05; range =1.23-13.54) and *Cyperus cuspidatus* (mean = 3.18±1.89; range =0.66-13.36) fall between these two.

There is no significant difference in charcoal width between *Arundinella setosa* and *Neyraudia reynaudiana* or *Cyperus cuspidatus*, nor between *Miscanthus sinensis* and *Neyraudia reynaudiana*. However, significant differences exist between the charcoal widths of all other vegetation types (ANOVA; P<0.05).

### 3.2.3 The area of charcoal

The charcoal size of the *Cryptomeria japonica* (mean = 47.91, range = 1.68-633.55), *Pinus massoniana* (mean = 29.97, range = 2.40-296.54), *Pinus yunnanensis* (mean = 26.78, range = 2.44-498.68) and *Pinus densata* (mean = 20.23, range = 0.08-315.99) are decreasing successively in the evergreen coniferous forests (Fig. 8). Only *Cryptomeria japonica* shows significant difference in charcoal area compared to other vegetation types (ANOVA; P<0.01). The charcoal area of the *Arundinella setosa* (mean = 23.79, range = 2.26-261.68), *Cyperus cuspidatus* (mean = 21.40, range = 2.42-233.66), *Neyraudia reynaudiana* (mean = 16.87, range = 0.44-205.53) and *Miscanthus sinensis* (mean = 10.86, range = 2.41-71.47) are decreasing successively in the warm-humid herbs. The significant difference of charcoal area only exists between *Miscanthus sinensis* and *Arundinella setosa* (ANOVA; P<0.01). The charcoal area distribution of the above two types is wide, with the maximum area up to 633.55 μm$^2$, but it still mainly concentrating between 0-100μm$^2$, and the percentage of content is more than 90%.



The charcoal area of the evergreen *Quercus glauca* forests and *Stipa capillata* grasslands is relatively small, mainly concentrated in the 0-50 $\mu m^2$, the proportions are 94% and 100%, respectively.

### 3.2.4 The length: width (L/W) ratio of charcoal

The range of charcoal L/W ratio of the four groups is relatively similar, and the
distribution ranges are also comparable. The proportions of charcoal distributed the ranges of the 1-2, 2-3, and >3 decreases sequentially. The warm-humid herbs show the largest variation range, with *Miscanthus sinensis* (mean = 1.55±0.47, range = 1.00-3.24) having the smallest average L/W ratio (Table 2) (Fig. 9). The average L/W ratio of *Neyraudia reynaudiana* (mean = 2.23±0.97, range = 1.01-5.44), *Arundinella setosa*
(mean = 1.95±0.96, range = 1.01-5.81), and *Cyperus cuspidatus* (mean = 1.92±0.86, range = 1.01-7.41) are decreasing sequentially. The percentages in the 1-2 are 83%, 50%, 67%, and 65%, respectively, followed by those in the 2-3 are 16%, 30%, 20%, and 26%. The proportion in the range of >3 is greater than 5%, with *Neyraudia reynaudiana* exceeding 10%. There is no significant difference between *Arundinella*
*setosa* and *Cyperus cuspidatus*, while significant differences exist among the L/W ratio of the other vegetation types (ANOVA; P<0.01).

The charcoal L/W ratio of *Pinus yunnanensis* (mean = 1.92±0.78, range = 1.00-4.90) is the greatest among the evergreen coniferous forests (Fig. 9). *Cryptomeria japonica* (mean = 1.73±0.63, range = 1.01-4.98), *Pinus massoniana* (mean = 1.76±0.78,
range = 1.00-6.19) and *Pinus densata* (mean = 1.71±0.77, range = 1.00-5.79) are close. The percentages in the 1-2 are 63%, 73%, 74%, and 78%, respectively, followed by those in the 2-3 are 28%, 22%, 21%, and 16%. The rest in the range of >3 is approximately 5%. Only the charcoal L/W ratio of *Pinus yunnanensis* shows a significant discrepancy compared to the *Pinus densata* (ANOVA; P<0.05).

The distribution ranges of evergreen *Quercus glauca* forests (mean = 1.68±0.64, range = 1.00-5.52) and *Stipa capillata* grasslands (mean = 1.75±0.69, range = 1.00-5.03)



are approach.

## 4 Discussion

In recent years, paleo-fires research has advanced significantly in both depth and breadth globally, establishing a robust framework for understanding fire dynamics-climate-vegetation-human activities interactions on centennial to millennial scales. These studies provide critical insights for predicting future fire regime dynamics. (Conedera et al.,2008; Andela et al.,2017; Miao et al.,2020; Ward et al.,2020; Petras and Istrate,2023). Current evidences suggests that the morphological traits of charcoal, as direct product of fires, are strongly influenced by factors, such as local vegetation structure, proximity to the fire source and wildfire type (Patterson et al.,1987; Ward and Hardy,1991; Vachula et al.,2022; Jensen,2007; Hudspith,2018; Flatley,2023). Regions with high vegetation diversity and coverage offer greater biomass availability for combustion, thereby enhancing charcoal production (Enache and Cumming,2007; Mustaphi and Pisaric,2014; Crawford,2016; Pereboom et al.,2020). Proximity to the fire source correlates with higher temperatures, promoting more complete biomass consumption, degree of char-coalification, and increased particles, leading to smaller charcoal (Clark and Hussey,1996; Vachula et al.,2021; Mindrescu et al.,2023). Wildfires are broadly classified into three types: crown fires, surface fires and ground fires (Scott,1989). Crown fires are distinguished by the high-intensity canopy combustion, elevated temperatures. Surface fires primarily consume litter and understory vegetation, generating lower temperatures and moderate charcoal yields. Ground fires are typically featured by slow burning, low temperature, which smolder organic soil layers, predominantly producing macroscopic charcoal (Hudspith et al.,2018; Feurdean et al.,2021). Charcoal formation is most prevalent in crown and surface fires, with crown fires contributing disproportionately due to their intense combustion dynamics and broader spatial impact.

In addition, the production of charcoal is further influenced by bark characteristics, including thickness, density, and surface structure, which modulate the burning speed



of woody plants (Da Silva et al., 2021; Nie Wen et al., 2021). As wildfires propagate

rapidly, the bark serves as a protective outer layer, with fire resistance. Hence, charcoal

generally originated from combustion of outer bark tissues (Bär et al., 2019).

Consequently, the morphological traits of charcoal derived from evergreen coniferous

forests dominated by large woody species are strongly associated with the structure

properties of the tree bark. The periderm, consisting of the phellem, phellogen and

phelloderm, displays a compact cellular arrangement with well-organized cell layers

(Fu, 2022). During pyrolysis and combustion, these structurally uniform cells fracture

under thermal stress, producing charcoal that is polygynous with clear and straight

edges. Quantitative analysis reveals that the average length, width, area, and L/W ratio

of charcoal from evergreen coniferous forests are 77.75 μm ,4.54 μm, 31.23 μm$^2$ and

1.78.

Charcoal morphological multiformity in evergreen coniferous forests

demonstrates interspecific differences, with *Cryptomeria japonica* taking on

particularly distinct characteristics. Comparative analyses uncover that *Cryptomeria*

*japonica* charcoal exhibits greater length, width and area measurements compared to

other species. This morphological divergence primarily stems from interspecific

variations in bark architecture and flammability properties.

The bark of *Cryptomeria japonica*, possesses a fine-textured, lightweight and

pliable structure (Editorial Committee of Flora of China, Chinese Academy of Sciences.

Flora of China. Science Press). However, Pinus species typically develop thicker,

coarser bark with higher resin content that enhances flammability (Editorial Committee

of Flora of China, Chinese Academy of Sciences. Flora of China. Science Press;

Pausas,2015a). Furthermore, *Cryptomeria japonica* employs an efficient sap-based

thermal regulation system during fires. The sap, circulating through xylem-phloem

interfacial tissues, facilitates evaporative cooling that mitigates burn severity (Bär A,

2019; Dukat, 2024). These combined factors, involving thinner bark and effective

physiological defenses, result in reduced combustion intensity and consequently larger

charcoal particle in *Cryptomeria japonica.* Among Pinus-dominated samples, obvious



discrepancy in charcoal length were observed between most species' pairs, except *Pinus yunnanensis* and *Pinus massoniana*. But no comparable variations were detected in width or area measurements. This dimensional discrepancy may reflect environmental variations across fire sites, such as burning temperature, wind velocity and fire duration, which impact charcoal formation process (Scott, 2000; Cochrane, 2009). All samples, originating from crown fires, suggests that bark structural differences constitute the primary determinant of charcoal morphological variation, while local fire conditions may explain secondary dimensional variations among pine species.

The charcoal derived from the evergreen *Quercus glauca* forests was generated exclusively through crown fires that only consumed outer bark tissues. Therefore, the charcoal morphology features are similar to those from the evergreen coniferous forests, including well-defined, smooth margins. However, the evergreen *Quercus glauca* forests are located in a high-altitude environment, without obvious tall arboreal trees, so the size of charcoal is smaller than that of evergreen coniferous forests, with average length, width, area and L/W ratio at 5.84 μm, 3.59 μm, 16.70 μm$^2$ and 1.68 in order.

The herbaceous leaves and stems were combusted completely, just leaving the charred basal stems, which achieved higher char-coalification degrees compared to woody species (Crawford and Belcher, 2014; Pereboom et al., 2020). The stems primarily consist of the epidermis, cortex, vascular cylinder, and pericycle. The epidermal cells are typically a single layer, with the outer cell walls often thickened by cornification. The cortical cells are loosely arranged, and the xylem in these plants is underdeveloped (Wang and Wang, 1989; Jiang et al., 2009; Fu, 2021). The relatively large pores within the stem contribute to the elongated morphology of the charcoal. The vascular bundles in the stem are arranged in a ring, and they break under external forces after thorough combustion, forming elongated charcoal with irregular margins. In *Stipa capillata* grasslands, these tufted grasses burn completely when exposed to wildfires, resulting in charcoal morphology that is mostly dot-like or granular, with serrated and irregular edges.

The average length, width, area and L/W ratio of grasslands are 5.84 μm、3.59 μm、



16.70 μm² and 1.68, while that of the warm-humid herbs are 6.26 μm, 3.49 μm、18.24 μm² and 1.91. The mostly charcoal morphology of both is resemble, with irregular

edges. But the size of warm-humid herbs is larger than those of grasslands. This is maybe the plants of warm-humid herbs are relatively tall, with robust stems, which are not completely burned during combustion. In addition, the variability of charcoal data from different vegetation types exhibits distinct discrepancy in the warm-humid herbs, and the main reason for this result might be the variety of plants. The robust stem was

not completely consumed, while other parts burned completely in different degrees, leading to a high char-coalification process.

Current research has emphasized L/W ratio of charcoal can be an effective proxy for distinguishing diverse fuel types (Umbanhowar and McGrath, 1998; Scott, 2010; Li et al., 2019; Hu et al., 2020; Vachula et al., 2021; Fuerdean et al., 2023). And this paper

provided that the L/W ratio of evergreen coniferous forests, evergreen *Quercus glauca* forests, *Stipa capillata* grasslands and warm-humid herbs are 1.78, 1.68, 1.75 and 1.91. Moreover, the distribution intervals of charcoal L/W ratio across various categories are quite similar, making it difficult to distinguish fuel types solely based on the charcoal L/W ratio, which contradicts conclusions from previous studies. The discrepancy may

arise because most of the previous studies were relied on laboratory simulation experiments (Umbanhowar and McGrath, 1998; Zhang and Lv, 2006; Fuerdean et al., 2023), where the controlled environment could not accurately replicate the conditions of natural wildfires. As a result, the L/W ratio of charcoal formed in wildfires differ from those produced in simulation experiments.

This study examined the morphological parameters from diverse vegetation types and found that the average length, width, and area of charcoal varied significantly among these types. ANOVA results further revealed prominent differences in the length, width, and area (ANOVA; $P<0.05$). these finding suggests that charcoal parameters, such as length, width, and area can be used as indicators to distinguish charcoal

originating from various vegetation types. To further explore the morphological and size differences of charcoal across vegetation types, this study employed a decision tree



model for classification, based on the correlations between charcoal parameters and the sample size (Fig. 10). Decision tree model is a widely used in classification due to its interpretability and effectiveness. The R programming language was utilized construct

the based on statistical data (length, width, area, and L/W ratio). The charcoal was categorized, with 70% of the sample allocated to the training set and 30% to the test set for model development. Given the varying influence of different charcoal parameters, including length, width, area and L/W ratio on classification performance, we calculated the relative contribution of each parameter to optimize model performance.

The model sets nodes to represent splitting conditions relied on variable thresholds, dividing branches into left and right according to these thresholds. Subsequent splits are made according to different conditions until classification is completed. The model demonstrates effective discrimination among three vegetation types: evergreen coniferous forests, evergreen *Quercus glauca* forests and *Stipa capillata* grasslands.

Ultimately, the model evaluation through cross-validation yielded a classification accuracy of 72.44%. To apply the model, newly measured charcoal data can be imported into R, enabling vegetation type prediction.

Charcoal, as a product of fires, is closely linked to vegetation type, bark structure, and wildfire intensity (Patterson et al., 1987; Ward and Hardy, 1991; Vachula et al.,

2022). Thicker and more compact bark tends to produce larger charcoal particles with smoother edges, while higher fire intensity leads to more complete combustion, resulting in smaller charcoal particles. Based on these discussions, charcoal is a complex product of fire events. When reconstructing paleo-fire history and past vegetation environments using sedimentary charcoal, a comprehensively analysis of

charcoal morphology, particle size, and concentration is essential for accurate environmental interpretation. This multi-proxy approach enhances the understanding of the interactions between fire regimes, vegetation composition, and climate.

## 5  Conclusions

This paper investigated charcoal samples from four distinct vegetation types under



the wildfires, and all the charcoal were extracted using a standardized pollen methodology, photographed under the microscope, measured using Image J for statistically analyzed of morphological characteristics. The charcoal morphology of evergreen coniferous forests is polygynous, flaky with flat edges and smooth surface, which is resemble to that of the evergreen *Quercus glauca* forests. The average length,

width, area and L/W ratio of evergreen coniferous forests are 5.84 μm, 3.59 μm, 16.70 μm$^2$ and 1.68 while the same parameters of the evergreen *Quercus glauca* forests are 7.75 μm, 4.54 μm, 31.23 μm$^2$ and 1.78. The charcoal characteristics from *Stipa capillata* grasslands are predominantly elongated with smaller sizes, displaying dot-shaped or granular, with irregular margins. Despite these particles share morphological

similarities with charcoal derived from warm-humid herbs, the charcoal size of warm-humid herbs presents larger dimensions and frequently occurs as bundled aggregates structures. The average length, width, area and L/W ratio of *Stipa capillata* grasslands are 3.32 μm, 1.98 μm, 4.53 μm$^2$ and 1.75. the same parameters of the warm-humid herbs are 6.26 μm, 3.49 μm, 18.24 μm$^2$ and 1.91. The classification accuracy of the

decision tree model achieved 72.44% in classifying these vegetation types. There are obvious differences in the morphology of charcoal produced by various vegetation types under the wildfire. Therefore, when using charcoal to interpret the environment formation, information about morphology characteristics, particle size, transport and deposition process of charcoal should be integrated, so as to systematically grasp the

complex relationship between natural wildfires and vegetation environment.





**Code and data availability**

All raw measurements of length, width, area and L/W are present in File S1. This dataset will be made available on request.

**Author contributions**

HS and XQL designed the study. HS, MJ and SRZ undertook the fieldwork, SRZ performed the experiments and wrote the original manuscript. HS and XYZ guided the structure and logic of the paper. All co-authors commented on the manuscript and provided input to the final version.

**Competing interests**

The authors declare that they have no known competing financial interests or personal relationships that could have appeared to influence the work reported in this paper.

**Disclaimer**

Publisher's note: Copernicus Publications remains neutral with regard to jurisdictional claims made in the text, published maps, institutional affiliations, or any other geographical representation in this paper. While Copernicus Publications makes every effort to include appropriate place names, the final responsibility lies with the authors.

**Acknowledgements**

We are deeply grateful to Jian Wang and Lili Lu for helpful discussions, and Huihai Wang for his advice with pollen extraction experiments.

**Financial support**

Our work was supported by the Youth Innovation Promotion Association of the Chinese Academy of Sciences (grant no. 2022071), the National Key Research and Development Program of China (grant no. 2022YFF0801502), Yunnan Revitalization Talent Support Program "Young Talent" Project (No. XDYC-QNRC-2022-0630) and General Program of Fundamental Research Programs of Yunnan Province (No. 202301AT070048).



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



**Figure 1.** Location map of charcoal samples (a: Sichuan Province, b: Yunnan Province, c: Guizhou Province, d: Inner Mongolia Autonomous Region; the photo of *Stipa capillata* was coming from *https://www.peopleapp.com/column/30039026390-500004021168*)

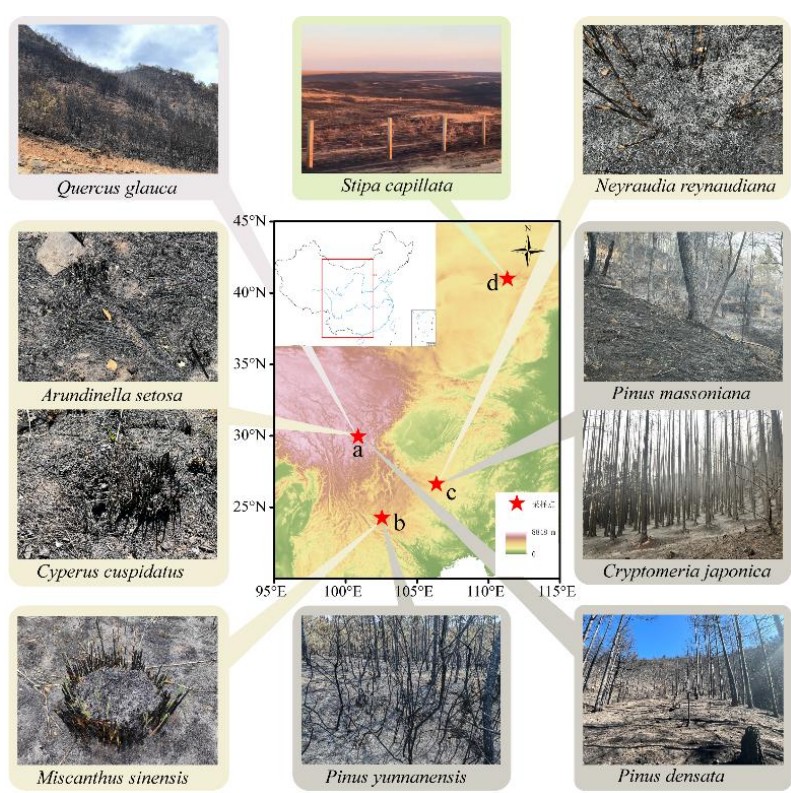





**Figure 2.** Charcoal of evergreen coniferous forests under microscope and SEM (1-4 represent respectively the forests of *Pinus yunnanensis, Cryptomeria japonica, Cryptomeria japonica* and *Pinus densata*)


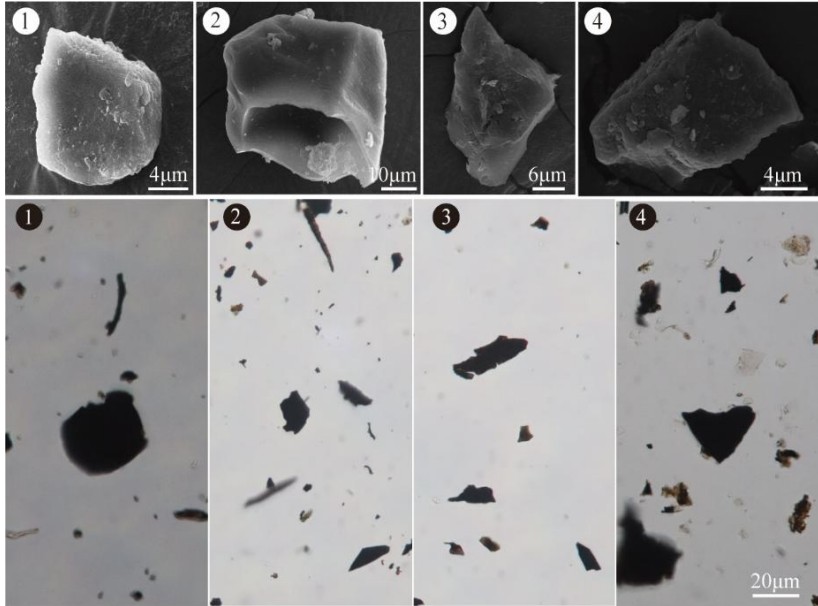





**Figure 3.** Charcoal of evergreen broad-leaf forests under microscope and SEM

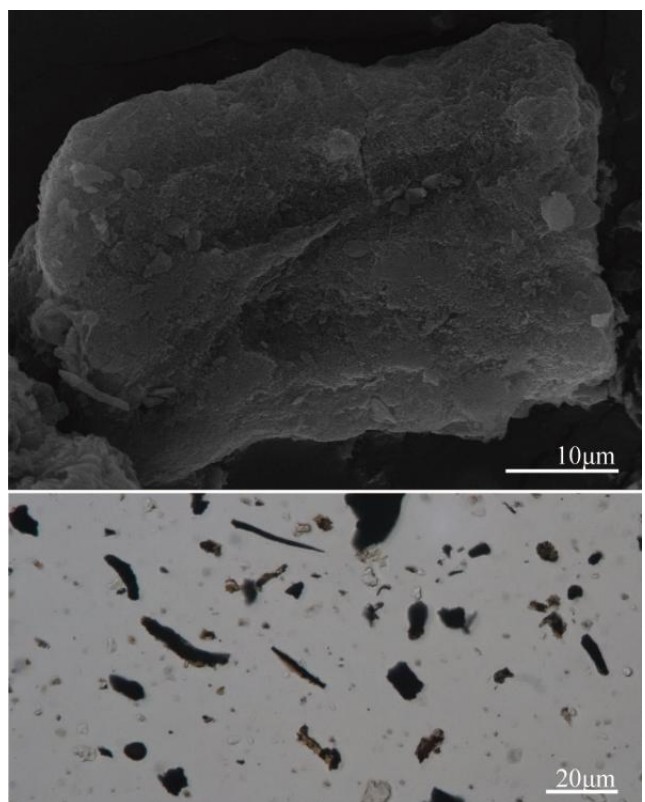





**Figure 4.** Charcoal of *Stipa capillata* grasslands under microscope and SEM

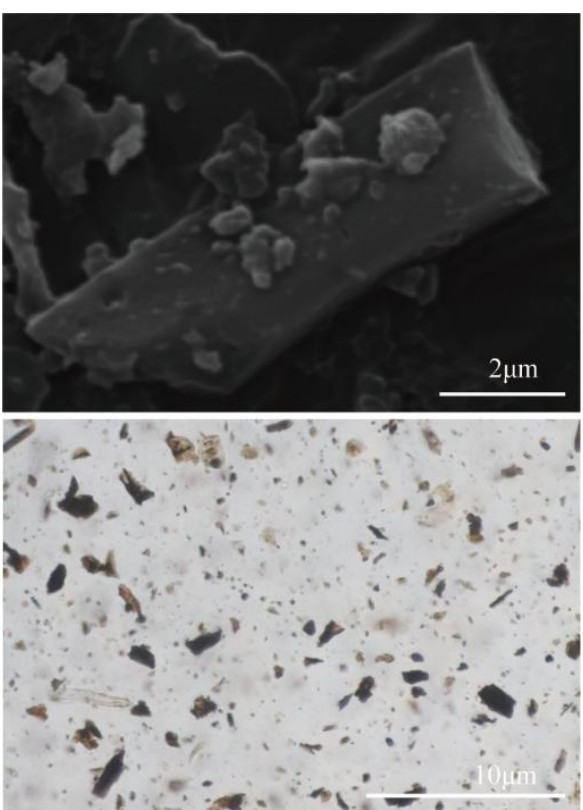







**Figure 5.** Charcoal of herbs under microscope and SEM (1-4 represent herbs of *Miscanthus sinensis*, *Neyraudia reynaudiana*, *Arundinella setosa* and *Cyperus cuspidatus*)


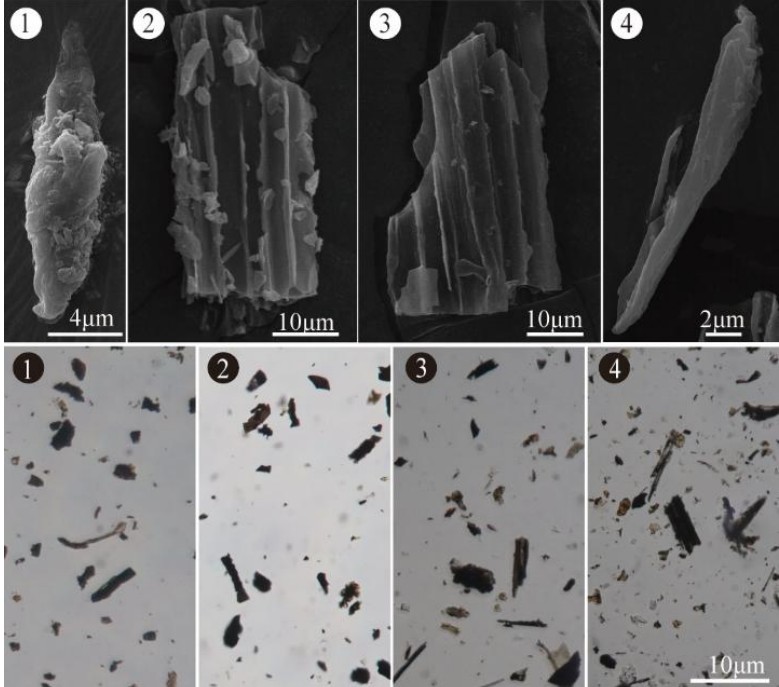







**Figure 6.** The box diagram showing the variation of charcoal length (a represents different dominant plant types; b represents different vegetation types)

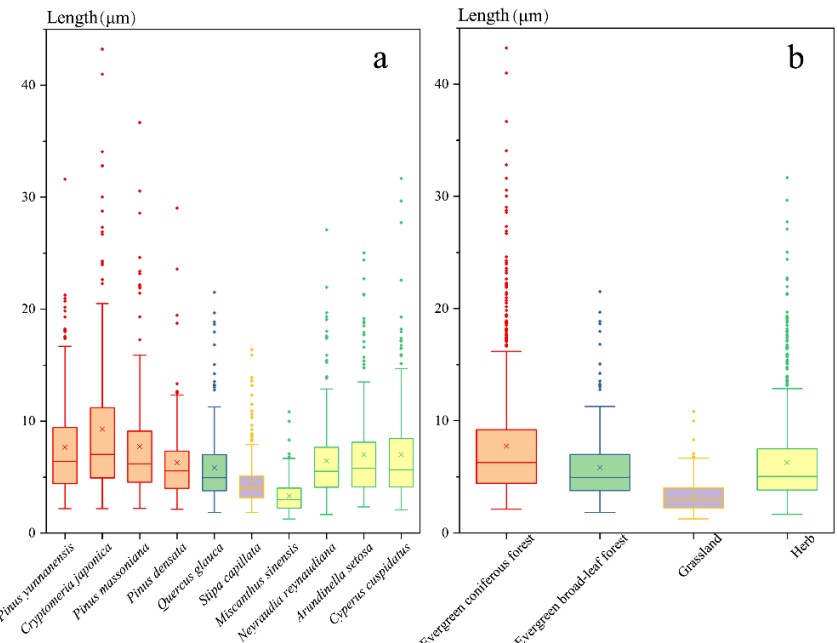







**Figure 7.** The box diagram showing the variation of charcoal width (a represents different dominant plant types; b represents different vegetation types)

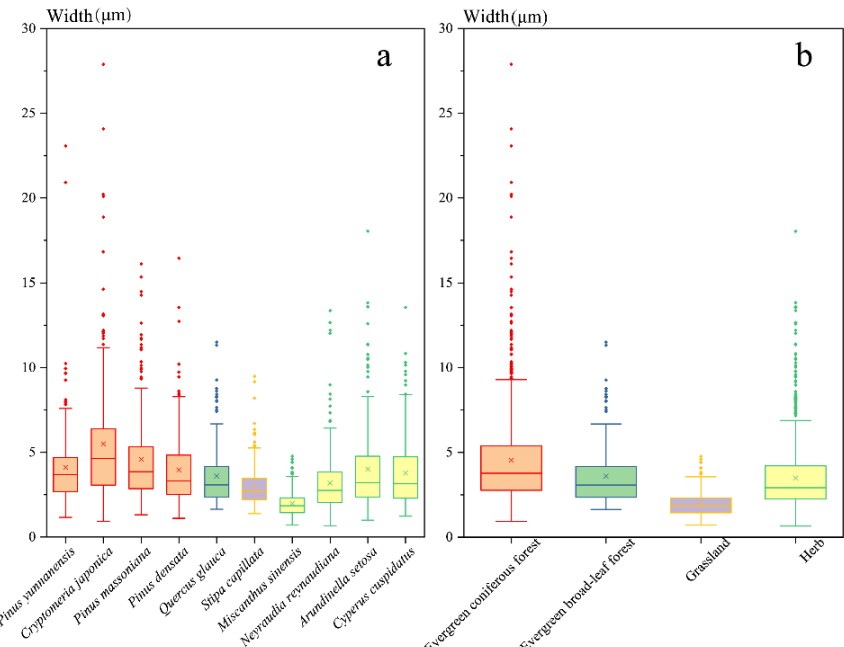






**Figure 8**. The box diagram showing the variation of charcoal area (a represents different dominant plant types; b represents different vegetation types)

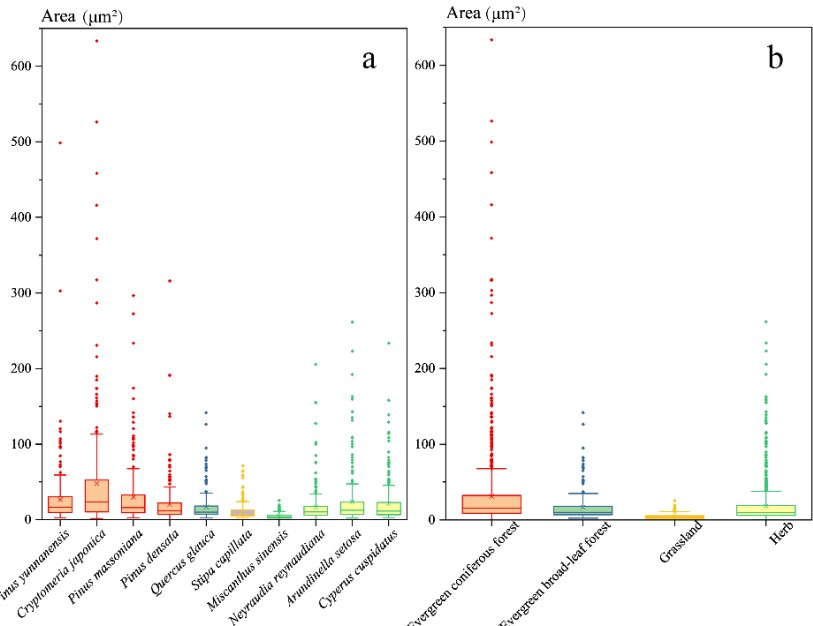







**Figure 9.** The box diagram showing the variation of charcoal L/W ratio (a represents different dominant plant types; b represents different vegetation types)

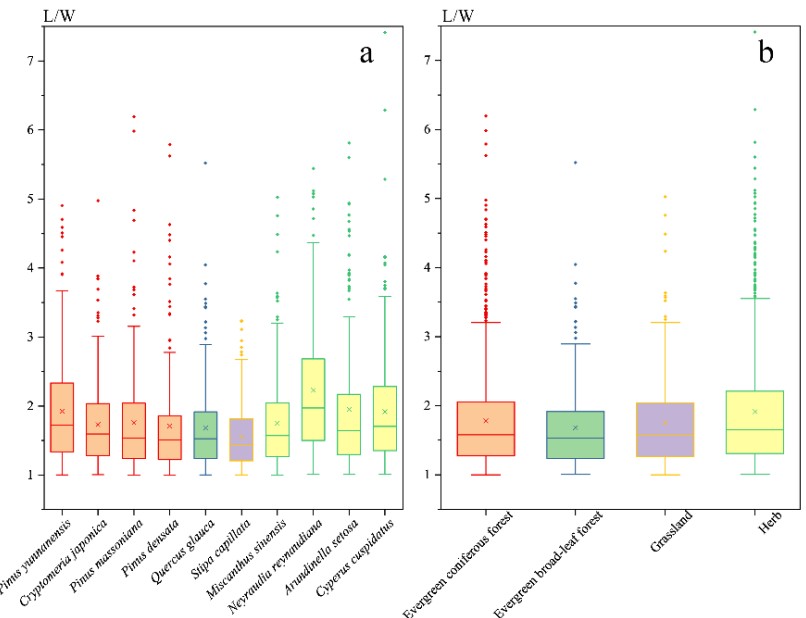







**Figure 10.** The diagram of decision tree model

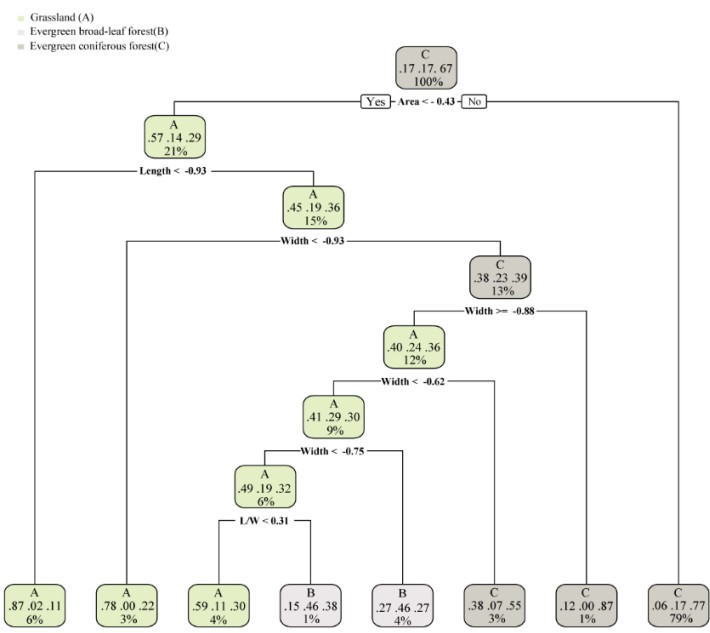








Table 1. Charcoal samples and classification

| Group | Dominant species |
|---|---|
| Evergreen coniferous forests | *Pinus yunnanensis*<br>*Cryptomeria japonica*<br>*Pinus massoniana*<br>*Pinus densata* |
| Evergreen broad-leaf forests | *Quercus glauca* |
| Grasslands | *Stipa capillata* |
| Herbs | *Miscanthus sinensis*<br>*Neyraudia reynaudiana*<br>*Arundinella setosa*<br>*Cyperus cuspidatus* |





Table 2. Parameters of charcoal

| Group | Dominant species | Length/μm | Width/μm | Area/μm² | L/W ratio |
|---|---|---|---|---|---|
| Evergreen coniferous forests | *Pinus yunnanensis* | 7.66±4.43 | 4.11±2.35 | 26.78 | 1.92±0.78 |
| | *Cryptomeria japonica* | 9.29±6.67 | 5.50±3.72 | 47.91 | 1.73±0.63 |
| | *Pinus massoniana* | 7.73±5.05 | 4.59±2.64 | 29.97 | 1.76±0.78 |
| | *Pinus densata* | 6.30±3.56 | 3.96±2.24 | 20.23 | 1.71±0.77 |
| Evergreen broad-leaf forests | *Quercus glauca* | 5.84±3.24 | 3.59±1.75 | 16.70 | 1.68±0.64 |
| Grasslands | *Stipa capillata* | 3.32±1.46 | 1.98±0.76 | 4.53 | 1.75±0.69 |
| Herbs | *Miscanthus sinensis* | 4.57±2.30 | 2.99±1.24 | 10.86 | 1.55±0.47 |
| | *Neyraudia reynaudiana* | 7.01±4.16 | 4.00±2.54 | 23.79 | 1.95±0.96 |
| | *Arundinella setosa* | 7.01±4.59 | 3.78±2.05 | 21.40 | 1.92±0.86 |
| | *Cyperus cuspidatus* | 6.46±3.74 | 3.18±1.89 | 16.87 | 2.23±0.97 |