# Peer review of "The charcoal morphology"

_EGUsphere, 2025_

## Referee Comment (RC1)

**General Comments**

Zhang et al. present results of a morphometric analysis of charcoal particles originating from wildfires in 10 different environments, each represented by one dominant vegetation species, and collectively covering 4 broader categories: coniferous forests, broadleaf forest, grassland, and other herbaceous ecosystems. One aim is to further test the idea that the length/width ratio of charcoal particles can indicate their source vegetation: an idea now quite widely employed in palaeo studies and generally thought to be well evidenced. Zhang et al. point out that such methods are normally justified based on simulations of wildfire, rather than real wildfires. Differences in size descriptors and in L/W ratio are identified between the different dominant species, and between the broader categories (i.e. conifers, broadleaf, grass, forbs), but the similarity of L/W distributions among the latter is held to be contrary to previous findings. The data is used to derive a classification tree (which relies largely on size rather than shape descriptors) and this is found to categorize the remaining data with 72% accuracy. It is suggested that this model may be used to categorize subsequently collected data according to fuel type.
* * *
The authors have collected a substantial amount of morphometric data on charcoal particles from ecosystems where such studies may be novel. In addition, they have made a fairly thorough survey of the relevant literature with which to contextualize their results (though there do appear to be occasional errors in the citations). However, the analysis presented here does not fully describe the work done, or justify the methods chosen, and the resulting claims to have improved upon previous understanding of the subject do not hold up.

First, a large amount of essential information is not provided. No information is given about the study areas, except to name the dominant vegetation species in each, without indication as to what proportion of the vegetation it accounts for. No information is given about the fires that formed the charcoal, except that they were 'natural wildfires' and 'crown fires' (and the latter is not possible in those ecosystems that lack tree cover). There are also many gaps in the description of the methods. In short, the contextual information needed for meaningful interpretation of charcoal morphology is absent. Meanwhile the authors' interpretations are frequently unclear, associating the morphometric data to statements about plant physiology and claims about the origin of the particles without demonstrating the connections.

The methods largely follow standard practice. However, the fitting of rectangles to the particle images in order to measure length and width may introduce substantial error (see specific comments for details). This might explain the lack of difference in L/W ratio seen between vegetation types, which the authors suggest undermines the current use of L/W ratio to indicate vegetation type. In any case, the most that the data presented here could do would be to show that such a method cannot be applied in the particular environments studied here. Yet the authors seem to suggest it refutes the method in general. (There is some ambiguity on this point, which stems from the sometimes imprecise wording.)

Regarding the decision tree employed, the accuracy with which a model trained on one part of a dataset can predict the other part of the same set gives no reliable indication of how well it would do with other data. It can only be assumed to apply to data sets subject to the same constraints – i.e. the same locations or strictly analogous ones, and the same sample processing, etc. Yet essential parameters of these data (such as selection of sampling locations and size fractions) are withheld from the reader. Most importantly, there is no attempt to show that the data collected can be considered representative of any broader categories, which is essential if the findings are to allow inferences about other data.

The ultimate conclusion, that the study has demonstrated morphometric differences between vegetation types, and that such data should not be interpreted in isolation from other environmental factors, is well-founded. However this only establishes that generally understood principles remain true in environments where they had not previously been tested, rather than advancing understanding of the field in general. The subject matter is well suited to *Biogeosciences*, but I think the interpretation of the data will need very thorough revision if it is to demonstrate a real advance in the subject.

**Specific Comments**

**Abstract**

Lines 42-43: *"playing a significant role in maintaining ecological balance"*

Wildfires can both maintain and disrupt ecological balances. They are not inherently a stabilizing process, so this should be reworded.

Lines 51-53:

> *"The findings indicate a gradual decrease in charcoal size across different ecosystems, such as evergreen coniferous forests, warm-humid herbs, evergreen broad-leaf forests, and grasslands."*

A "gradual decrease" implies some order to the size distributions, and implicitly some underlying factor to which size is correlated, and with respect to which the change can be 'gradual'. But here it seems the ecosystems have been placed in order of decreasing charcoal size: the actual claim is only that size differs between the ecosystems. The phrase 'such as' should not be used as it implies that the named ecosystems are only a sample of those being referred to.

**Introduction**

Lines 90-91:

> *"Charcoal, as a direct product of fire, possesses unique properties such as high-temperature resistance, ease of preservation, high yield and widespread distribution."*

This needs rewording as it is not clear exactly what is meant. Does charcoal have "high-temperature resistance" when it burns easily? To say it has "high yield" is neither true or false: it depends on the conditions and on what it is being compared to. I think the intended point here is that charcoal is generally preserved in some form after wildfire, whereas other products and effects of fire are not.

Line 100: *"Being chemically and biologically inert..."*

As above, this is a matter of comparison with other materials, and charcoal can be destroyed by chemical and biochemical processes, so the wording should be modified.

Lines 113-116:

> *"Umbanhowar and McGrath (1998) observed distinct characteristics between herbaceous and woody charcoal, noting that herbaceous charcoal tends to be elongated with length and area are 563 μm and 5630 μm², while woody charcoal typically appears square or round, measuring 380 μm and 64946 μm² in length and area."*

The way this is written implies that Umbanhowar & McGrath made exact and universal claims about sizes of charcoal particles. The measurements refer to the particular size fraction they studied, and do not seem necessary here.

Lines 132-135:

> *"While previous studies have relied on laboratory simulation experiments, which cannot fully replicate the burning conditions of actual wildfires, there exists a notable gap in systematic research on the charcoal morphology arising from various plants under natural wildfire scenarios."*

It is true that work on charcoal morphology has mostly used either experimentally produced charcoal from furnaces, bonfires, etc., or palaeocharcoal from sediment deposits. However, the purpose of such work is normally to interpret the latter, which will be shaped by both the characteristics of the fire and by taphonomic processes: transportation, burial, chemical alteration within the deposit, etc. It isn't clear that there is some unique value in measuring the shape and size of particles after the fire, but before any taphonomic alteration. In addition, the lack of information on the fires means that the reader cannot assume that there has not already been taphonomic alteration of the charcoal studied here. Claiming the importance of "natural wildfire scenarios" makes little sense when no information on the fires is offered.

Lines 138-140:

> *"Despite observed differences in L/W ratio of charcoal from distinct plants, specific thresholds for classification have yet to be established. Therefore, further extensive research is warranted in this area."*

The idea that there should be specific thresholds to separate plant types is questionable, since it is established that L/W ratio can be affected not only by plant type but by characteristics of the burn and of taphonomic processes. In fact, Umbanhowar et al. (2006)[1] came close to defining a threshold, claiming that grasses typically produce median L/W ≥ 3.5, and deciduous trees 2.0 to 3.0. Aleman et al. (2013)[2] suggested a L/W threshold of 2.0. But any such threshold must be considered dependent on similarity of fire conditions and post-fire alterations.
* * *
1: Umbanhowar et al. (2006) Asymmetric vegetation responses to mid-Holocene aridity at the prairie–forest ecotone in south-central Minnesota. *Quaternary Research*, vol. 66, pp. 53-66.
2: Aleman et al. (2013) Tracking land-cover changes with sedimentary charcoal in the Afrotropics. *The Holocene* 23(12), 1853–1862.

**Materials and methods**

No information is given about the fires from which the charcoal originates. This is absolutely essential. When did the fires take place, and how long after this were the samples collected? What were the weather conditions, which may have affected the charcoal between formation and collection? What is known of the fires' causes? Were any measurements or observations of the fires available – e.g. their duration, extent, flame length, etc.? Were any of them subject to control or containment, or were they allowed to burn naturally? The intensity and severity of the fires affect charcoal properties, so we must know about them. If the charcoal was not collected immediately, were any immediate post-fire observations made?

Line 154: *"a total of 10 charcoal samples were collected..."*

How were the samples collected? How was each location chosen, and was the whole sample taken at a single point at each? They were presumably taken from the ground, not from trunks etc.? Were they taken from the surface over a defined area, or between the surface and a certain depth at one point? Was there any discrimination as to particle size? For example, if larger charred pieces were present, were they discarded, or broken up to fit the containers? Were areas of 'pure' charcoal and ash sought out, or was there also soil etc. in the samples?

Lines 155-158:

> *"...which were classified into three groups according to vegetation types, including 4 evergreen coniferous forests, 1 evergreen broad-leaf forest, and 1 grassland. In addition, 4 warm-humid herbs were also included"*

This is four groups. (Or is there something different about the herbaceous samples that means they are counted separately from the grouping of three vegetation types?)

Lines 159-174:

There is no reference to the samples being sieved but the process begins with (presumably) untreated charcoal, collected (presumably) from the forest floor, and ends with particles that can be imaged at 400x magnification. How large was the macroscopic charcoal, and what happened to it?

Line 161: *"hydrochloric acid (HCl)"*

What concentration?

Line 163: *"After a period of reaction time..."*

How long?

Lines 168-169: *"A specific amount of water and glacial acetic acid was added to the beakers..."*

How much?

Lines 170-171: *"heated in a water bath with a mixture of sulfuric acid and acetic anhydride"*

This needs detail added on temperature, and amount and concentration of reagents.

Lines 173-174:

> *"Each sample was counted and photographed under the light microscope at 400X magnification (Zeiss AxioImager.A2)."*

How were the samples distributed for counting and imaging? The extraction process ends with dried samples, according to the text above. Were the samples dried onto (e.g.) glass slides and then imaged, or (as I would assume) distributed in water in some sort of counting chamber?

What lighting was used, and was this the same for both counting and imaging? Was there any defined process to differentiate charcoal from other blackish material that may have been present?

Lines 181-182:

> *"Select the charcoal to be measured and use the command Edit—Selection—Fit Rectangle to determine the length and width of the samples."*

Why was this method used instead of the 'Analyze particles' function? Image J fits rectangles according to the orientation of the image, so the orientation of the particle with respect to the image will affect the result: only a particle oriented perpendicular to the the image will be accurately measured. 'Analyze particles' uses a fitted ellipse, which is independent of the image orientation. I would expect this difference to have a substantial effect on results.

Lines 183-184:

> *"Additionally, charcoal was selected under the microscope and transferred to the stage using appropriate tools."*

This appears to refer to the samples to be imaged with SEM as described in the next sentence. The 'stage' may refer to an SEM specimen stub? What were the 'appropriate tools'?

***Results***

Line 190: *"The evergreen coniferous forests are divided into four groups..."*

I think this refers to the four different forests from which samples were taken, but the wording makes it sound like a subsequent treatment of the data.

Line 192-197:

> *"The undergrowth is accompanied by some small shrubs, and the surface debris is mainly covered with fallen leaves and a few small dry branches. Charcoal deposited on the ground surface after wildfire is primarily composed of remnants plant tissues and undergrowth debris, involving leaves, bark and branches. Only the bark of the trunks underwent combustion, while the internal secondary phloem and xylem remained unaffected."*

These appear to be the results of pre-fire and post-fire surveys, which are not referred to in the methods. The source of the information needs to be explained. Assuming that the fires were entirely natural, there would have been no pre-fire survey, so perhaps the first sentence refers to another site analogous to the four burned areas? Is the description intended to apply to all four coniferous sites, and were there no differences between them?

Line 198 (and subsequently): *"polygynous"*

I think a different word is intended here. The botanical use of this word can only refer to the flowering parts of a plant.

Lines 206-207: *"with clear and smooth edges"*

I cannot tell what is meant by 'clear' in this context.

Line 213: *"And the level of char coalification process was heavily higher..."*

The meaning of 'higher' is not clear. (And 'charcoalification' should be one word.)

Line 219: *"characterized by relatively tall plants"*

Please indicate the actual height.

Lines 220-221: *"it underwent complete combustion, only leaving charred, sturdy stems behind..."*

This is a contradiction; complete combustion implies no char remaining.

Lines 221-222: *"This process results in a high degree of char-coalification."*

The meaning is unclear. This could refer to a large amount of the pre-burn biomass remaining as char, but that contradicts the previous line.

Lines 239-241 (and throughout § 3.2):

> *"Statistical analysis reveals no great difference in the length of charcoal between* Pinus yunnanensis *and* Pinus massoniana, *whereas the discrepancy exists among the other species (ANOVA; P<0.05)."*

Please provide full details, including specific p-values rather than if above/below 0.05.

Lines 300-301:

> *"The proportions of charcoal distributed the ranges of the 1-2, 2-3, and >3 decreases sequentially."*

I cannot understand this. There are differences in the L/W ratio, but in what way do they form a sequence?

**Discussion**

Lines 332-334:

> *"Regions with high vegetation diversity and coverage offer greater biomass availability for combustion…"*

Diversity doesn't seem relevant to this point; and while greater biomass potentially means more charcoal produced, the point is not connected to charcoal morphology.

Lines 335-338:

> *"Proximity to the fire source correlates with higher temperatures, promoting more complete biomass consumption, degree of char-coalification, and increased particles, leading to smaller charcoal"*

I find this statement quite muddled. I don't think 'fire source' refers to ignition point, so the claim seems to relate to distance from the area burned (in the sense of the area in which flaming combustion occurs); but the great majority of charcoal will originate from within that area. The authors also need to clarify what is meant by 'degree of charcoalification', and explain the link to greater fragmentation of charcoal.

Lines 345-347:

> *"Charcoal formation is most prevalent in crown and surface fires, with crown fires contributing disproportionately due to their intense combustion dynamics and broader spatial impact."*

It is not clear what 'broader spatial impact' means. The statement needs to be supported with references and/or mechanistic explanation.

Lines 350-352:

> *"As wildfires propagate rapidly, the bark serves as a protective outer layer, with fire resistance. Hence, charcoal generally originated from combustion of outer bark tissues (Bär et al., 2019)."*

Bark does not protect leaves, flowers, etc., or litter. Plants that have no bark also form charcoal. Some charcoal assemblages may be mainly bark charcoal, but it is not typical. The statement is not supported by the citation, which does not mention charcoal at all.

Lines 353-355:

> *"Consequently, the morphological traits of charcoal derived from evergreen coniferous forests dominated by large woody species are strongly associated with the structure properties of the tree bark."*

This claim is not evidenced. It is not demonstrated that the charcoal originates from bark, or that bark should produce the particular sizes and morphologies seen.

Lines 376-378:

> *"These combined factors, involving thinner bark and effective physiological defenses, result in reduced combustion intensity and consequently larger charcoal particle in* Cryptomeria japonica.*"*

Why does thin bark reduce combustion intensity? Why does lower intensity lead to larger particles? More generally, the attribution of reduced intensity to the physiology of the tree at this scale is implausible: perhaps the features of the bark result in lower intensity if only the bark burns, but the heating it is subjected to will be determined by the intensity with which the entire surrounding fuel complex burns.

Lines 378-379:

> *"Among Pinus-dominated samples, obvious discrepancy in charcoal length were observed between most species' pairs…"*

This is not obvious from Figure 6, even if p < 0.05 is met. Distributions for all 4 *Pinus* species look fairly similar.

Lines 380-383:

> *"But no comparable variations were detected in width or area measurements. This dimensional discrepancy may reflect environmental variations across fire sites, such as burning temperature, wind velocity and fire duration, which impact charcoal formation process (Scott, 2000; Cochrane, 2009)."*

The lengths of particle images are highly unlikely to differ if both the widths and areas do not differ. Perhaps the significance threshold is just met for length (though not in all pairings) and just missed

for width and area? Without the full statistical results we don't know. Some 'environmental variations' are offered as causes of this result, but no mechanisms are suggested to explain this.

Lines 383-385:

> "*All samples, originating from crown fires, suggests that bark structural differences constitute the primary determinant of charcoal morphological variation...*"

The claim that all the samples come from crown fires is not made prior to this point. It obviously cannot apply to the grassland and herbaceous samples, and for the forest samples must surely mean surface fire + crown fire. The complete lack of information on the fires themselves is again a problem here, but I take this to mean that all the forest fires included crown fires. It does not follow from that that all the charcoal originates from bark.

Lines 387-388:

> "*The charcoal derived from the evergreen* Quercus glauca *forests was generated exclusively through crown fires that only consumed outer bark tissues.*"

This is not possible. I think the intended claim is that the majority of the charcoal was bark charcoal, but this must be evidenced.

Lines 390-392:

> "*the evergreen* Quercus glauca *forests are located in a high-altitude environment, without obvious tall arboreal trees, so the size of charcoal is smaller than that of evergreen coniferous forests*"

This seems like a non-sequitur. Is it the altitude or the canopy height that affects charcoal size, and how?

Lines 395-396:

> "*charred basal stems, which achieved higher char-coalification degrees compared to woody species (Crawford and Belcher, 2014; Pereboom et al., 2020).*"

It isn't clear what is meant by a higher degree of charcoalification, and neither of the references helps. I suspect the intended meaning is degree of fragmentation – i.e. a higher degree meaning smaller particles?

Lines 400-401:

> "*The relatively large pores within the stem contribute to the elongated morphology of the charcoal.*"

No explanation is given for how this occurs.

Line 404: *"these tufted grasses burn completely when exposed to wildfires"*

Complete combustion would mean no charcoal left to analyze.

Lines 415-416:

> *"other parts burned completely in different degrees, leading to a high char-coalification process."*

The meaning of 'high charcoalification process' is unclear.

Lines 419-421:

> *"And this paper provided that the L/W ratio of evergreen coniferous forests, evergreen* Quercus glauca *forests,* Stipa capillata *grasslands and warm-humid herbs are 1.78, 1.68, 1.75 and 1.91."*

These are presumably the median or mean values, but there is no attempt to demonstrate that they are generally representative of these environments (e.g. though randomised sampling) or to estimate confidence intervals, so their interpretive value is limited.

Lines 422-424:

> *"the distribution intervals of charcoal L/W ratio across various categories are quite similar, making it difficult to distinguish fuel types solely based on the charcoal L/W ratio, which contradicts conclusions from previous studies."*

It would be more accurate to say 'contrasts with previous studies'. Firstly, I do not think any cited author has claimed that fuel type can be determined from charcoal L/W ratio alone; the method is always used alongside others, and interpretations are environment-specific. If such a claim has been made, it should be cited here. Relatedly, the findings here are valid only for the environments studied (if at all), and cannot invalidate those of (e.g.) Umbanhowar & McGrath (1998) in the environments they studied. Finally, as described above, the method of measuring L/W by fitting rectangles will tend to underestimate variation, which may partly explain the lack of variation seen here.

Lines 433-435:

> *"these finding suggests that charcoal parameters, such as length, width, and area can be used as indicators to distinguish charcoal originating from various vegetation types."*

This is already known, so it would be better to say "these findings provide further evidence..."

Lines 439-447:

> *"The R programming language was utilized…*
> *...Subsequent splits are made according to different conditions until classification is*
> *completed."*

All this text should be in the Methods.

Lines 450-452:

> *"Ultimately, the model evaluation through cross-validation yielded a classification accuracy*
> *of 72.44%. To apply the model, newly measured charcoal data can be imported into R,*
> *enabling vegetation type prediction."*

The claim being made here needs to be clarified.  If the model allows charcoal measurements to be converted to vegetation types (at 72% accuracy), under what circumstances is it valid?  How can the authors demonstrate that it is applicable beyond their own data set?  Since only around 5% of particles are ultimately classified according to shape, the model relies mainly on particle size; therefore the size fraction used will be essential, but this has not been explained.

Lines 455-457:

> *"Thicker and more compact bark tends to produce larger charcoal particles with*
> *smoother edges, while higher fire intensity leads to more complete combustion,*
> *resulting in smaller charcoal particles."*

These statements remain unevidenced.

**Conclusions**

Lines 480-484:

> *"there are obvious differences in the morphology of charcoal produced by various*
> *vegetation types under the wildfire. Therefore, when using charcoal to interpret the*
> *environment formation, information about morphology characteristics, particle size,*
> *transport and deposition process of charcoal should be integrated…"*

This was already understood, and yet it is the only generalized statement in the Conclusions (the rest being a restatement of the methods and results).

If the authors stand by their claim to have undermined the use of L/W ratio as an indicator of vegetation type, and to have produced a superior method in their decision tree model, they should restate that here.  As I don't think those claims are defensible, the more moderate conclusion would be that the L/W ratio did not differentiate vegetation types in this case.  As it is specifically grasses (or Poales) that are known to produce elongated particles, perhaps it is *Miscanthus sinensis* that is unusual?

**Figures**

Figs 2-5:

There is no reference to the SEM images in the results or discussion, so they seem unnecessary.

Figs 6-9:

Please explain exactly what is shown, as there are different conventions for box plots.  How are the outliers defined?  Does 'X' indicate the mean?

Fig 10:

The diagram needs to be explained in the caption.  E.g., why are categories assigned to the root node and internal nodes?

**Technical Corrections**

Line 175:

Please provide a reference for ImageJ, with details of the version used.

Line 213 (and subsequently):

'Charcoalification' should be a single word, without a hyphen.

Line 335:

"Mustaphi and Pisaric,2014"  should be "Courtney Mustaphi and Pisaric, 2014".  (It is given correctly in the reference list, though still under 'M'.)

Line 335:

The citation "Crawford,2016" is not present in the reference list.

Reference list:

Crawford & Belcher (2014):  The wrong journal is stated.

Zhang & Lv (2006):   This should be Zhang & Lu.  (The same error is made in the text.)

---

## Author Comment (AC2)

General Comments:

R: Thank you for pointing these comments. The data analysis and methodological validation for this study will be supplemented in the revised manuscript. And we also add some information into the "materials and methods" section. The method of measuring the charcoal particles and the choice of the decision tree model will also be explained in detail in the revised manuscript.

Specific comments:

Abstract:

Lines 42-43:" Wildfires can both maintain and disrupt ecological balances. They are not inherently a stabilizing process, so this should be reworded."

R: Thanks, we will follow the suggestions and make the according changes.

Lines 51-53:" A "gradual decrease" implies some order to the size distributions, and implicitly some underlying factor to which size is correlated, and with respect to which the change can be 'gradual'. But here it seems the ecosystems have been placed in order of decreasing charcoal size: the actual claim is only that size differs between the ecosystems. The phrase 'such as' should not be used as it implies that the named ecosystems are only a sample of those being referred to."

R: Thank you for pointing this, we will clarify that.

Introduction:

Lines 90-91:" This needs rewording as it is not clear exactly what is meant. Does charcoal have "high temperature resistance" when it burns easily? To say it has "high yield" is neither true or false: it depends on the conditions and on what it is being compared to. I think the intended point here is that charcoal is generally preserved in some form after wildfire, whereas other products and effects of fire are not."

R: Thanks, we will revise this part in the revised manuscript.

Lines 100:" As above, this is a matter of comparison with other materials, and charcoal can

be destroyed by chemical and biochemical processes, so the wording should be modified."

R: Thank you for this suggestion, we will rephrase in the revised manuscript.

Lines 113-116:" The way this is written implies that Umbanhowar & McGrath made exact and universal claims about sizes of charcoal particles. The measurements refer to the particular size fraction they studied, and do not seem necessary here."

R: Thank you for this suggestion, we revised this sentence based on your suggestion.

Lines 132-135:" It is true that work on charcoal morphology has mostly used either experimentally produced charcoal from furnaces, bonfires, etc., or palaeocharcoal from sediment deposits. However, the purpose of such work is normally to interpret the latter, which will be shaped by both the characteristics of the fire and by taphonomic processes: transportation, burial, chemical alteration within the deposit, etc. It isn't clear that there is some unique value in measuring the shape and size of particles after the fire, but before any taphonomic alteration. In addition, the lack of information on the fires means that the reader cannot assume that there has not already been taphonomic alteration of the charcoal studied here. Claiming the importance of "natural wildfire scenarios" makes little sense when no information on the fires is offered."

R: Thank you for pointing these. Current studies about simulation experiments demonstrate the significant value of charcoal morphology in distinguishing fuel types. However, researches on charcoal burning from natural wildfires remain limited and the simulation experiments cannot fully replicate the complex conditions of actual wildfires. Therefore, the study of charcoal burning from natural wildfires without taphonomic processes is crucial for summarizing the morphological characteristics of charcoal from different vegetation types.

Lines 138-140:" The idea that there should be specific thresholds to separate plant types is questionable, since it is established that L/W ratio can be affected not only by plant type but by characteristics of the burn and of taphonomic processes. In fact,

Umbanhowar et al. (2006)[1] came close to defining a threshold, claiming that grasses typically produce median L/W ≥ 3.5, and deciduous trees 2.0 to 3.0. Aleman et al. (2013)[2] suggested a L/W threshold of 2.0. But any such threshold must be considered dependent on similarity of fire conditions and post-fire alterations.'

R: Thank you for the sources. We will cite them and rephrase in the revised manuscript.

Materials and methods:

Lines 154:" How were the samples collected? How was each location chosen, and was the whole sample taken at a single point at each? They were presumably taken from the ground, not from trunks etc.? Were they taken from the surface over a defined area, or between the surface and a certain depth at one point? Was there any discrimination as to particle size? For example, if larger charred pieces were present, were they discarded, or broken up to fit the containers? Were areas of 'pure' charcoal and ash sought out, or was there also soil etc. in the samples?"

R: Thank you for the specific and helpful suggestions. We will extend the description according to your comments in an updated version of the manuscript.

Lines 155-158:" This is four groups. (Or is there something different about the herbaceous samples that means they are counted separately from the grouping of three vegetation types?)"

R: Thank you for this comment. According to the vegetation types, this is actually four groups. The warn-humid herbs are not the classical vegetation type, which is great different from grassland and the model mentioned in the discussion mainly involves three vegetation types. So, we choose this way to descript that.

Lines 159-174:" There is no reference to the samples being sieved but the process begins with (presumably) untreated charcoal, collected (presumably) from the forest floor, and ends with particles that can be imaged at 400x magnification. How large was the macroscopic charcoal, and what happened to it?"

R: Thank you for your comments. We will incorporate the mentioned suggestions into our

revised manuscript.

Lines 181-182:" Why was this method used instead of the 'Analyze particles' function? Image J fits rectangles according to the orientation of the image, so the orientation of the particle with respect to the image will affect the result: only a particle oriented perpendicular to the image will be accurately measured. 'Analyze particles' uses a fitted ellipse, which is independent of the image orientation. I would expect this difference to have a substantial effect on results."

R: Thank you for your remark. We only use the "Analysis particles" function of image J to measure the area of charcoal. It was found through repeated verification that the fit rectangle can measure the length and width of charcoal correctly, irrespective of particles orientation. While the fit ellipse can determine the major and minor axes of the ellipse with an equivalent area, it cannot measure the actual length and width of charcoal. The dimensions (length and width) of the bounding rectangle vary depending on the particles's orientation within the image.

Lines 183-184:" This appears to refer to the samples to be imaged with SEM as described in the next sentence. The 'stage' may refer to an SEM specimen stub? What were the 'appropriate tools'?"

R: Yes, the "stage" is the SEM specimen stub, and the appropriate tool I used is hair.

Results:

Lines 190:" I think this refers to the four different forests from which samples were taken, but the wording makes it sound like a subsequent treatment of the data."

R: Thanks, we will clarify that.

Lines 192-197:" These appear to be the results of pre-fire and post-fire surveys, which are not referred to in the methods. The source of the information needs to be explained. Assuming that the fires were entirely natural, there would have been no pre-fire survey, so perhaps the first sentence refers to another site analogous to the four burned areas?

Is the description intended to apply to all four coniferous sites, and were there no differences between them? "

R: Thanks for your comments. All samples discussed in this study were collected from natural wildfires. And the descriptions of fuel types are based on field observations conducted in both the burned area and its immediate surroundings.

Lines 198:" I think a different word is intended here. The botanical use of this word can only refer to the flowering parts of a plant."

R: Thanks, we will clarify in the revised manuscript.

Lines 206-207:" I cannot tell what is meant by 'clear' in this context."

R: Thank you for highlighting this, we will change.

Lines 213:" The meaning of 'higher' is not clear. (And 'charcoalification' should be one word.)"

R: Thanks, we will correct the expression and modify the "char-coalification" to be one word in the revised manuscript.

Lines 219:" Please indicate the actual height."

R: We will add some information about this according to your comment.

Lines220-221:" This is a contradiction; complete combustion implies no char remaining"

R: Thanks, we will rewrite this sentence.

Lines 221-222:" This is a contradiction; complete combustion implies no char remaining."

R: We will revise the sentence to make the expression more reasonable.

Lines239-241:" Please provide full details, including specific p-values rather than if above/below 0.05."

R: Thank you for your helpful suggestions. We will provide more detailed information about

the statistical analysis in the revised manuscript.

Lines 300-301:" I cannot understand this. There are differences in the L/W ratio, but in what way do they form a sequence?"

R: Thank you for the comment. This part we want to show is that the proportion of L/W distributed across different intervals, including 1-2, 2-3, and >3, gradually decreases, rather than forming a sequence. We want to analyze the proportion of L/W in different intervals to determine in which interval it is primarily concentrated. This approach may help us identify the threshold of L/W for distinguishing different vegetation types.

Discussion:

Lines 332-334:" Diversity doesn't seem relevant to this point; and while greater biomass potentially means more charcoal produced, the point is not connected to charcoal morphology."

R: Thanks for the remark. We will rewrite this sentence. This part wants to show that the morphological characteristics of charcoal reflect the vegetation assemblage rather than a single plant species. A great diversity of vegetation types provides more combustible biomass, which affects both the diversity of charcoal morphologies and the charcoal yield.

Lines 335-338:" I find this statement quite muddled. I don't think 'fire source' refers to ignition point, so the claim seems to relate to distance from the area burned (in the sense of the area in which flaming combustion occurs); but the great majority of charcoal will originate from within that area. The authors also need to clarify what is meant by 'degree of charcoalification', and explain the link to greater fragmentation of charcoal."

R: Thank you for these suggestions. Regarding your first point, we will check and make the sentences reasonable. Scott (2010) has addressed this and provide a clear definition for charcoalification. And we will also explain the relationship between the "degree of charcoalification" and "greater fragmentation of charcoal".

Lines 345-347:" It is not clear what 'broader spatial impact' means. The statement needs to be supported with references and/or mechanistic explanation."

R: Thanks for pointing us to that. We will change another expression to make it clear and provide references.

Lines 350-352:" Bark does not protect leaves, flowers, etc., or litter. Plants that have no bark also form charcoal. Some charcoal assemblages may be mainly bark charcoal, but it is not typical. The statement is not supported by the citation, which does not mention charcoal at all."

R: Thanks. We agree that bark does not protect leaves, flowers, etc., or litter. We want to express that in vegetation types dominated by large plants, such as coniferous forests, the burning process is generally limited to the bark, preserving the internal structure. We will reposition this citation and supplement it with additional references for further support.

Lines 353-355:" This claim is not evidenced. It is not demonstrated that the charcoal originates from bark, or that bark should produce the particular sizes and morphologies seen."

R: Thanks. As mentioned previously, charcoal can originate not only from bark, but also from leaves, litter and other plant materials. Tet, existing researches show little morphological distinction between charcoal from leaves and twigs. This leads us to argue that variations in charcoal size are primarily caused by differences in bark structure.

Lines 376-378:" Why does thin bark reduce combustion intensity? Why does lower intensity lead to larger particles? More generally, the attribution of reduced intensity to the physiology of the tree at this scale is implausible: perhaps the features of the bark result in lower intensity if only the bark burns, but the heating it is subjected to will be determined by the intensity with which the entire surrounding fuel complex burns."

R: Thank for this remark. Because the sap can lower the temperature on the surface trees.

Thinner bark allows the sap to perform its protective function more effectively. A lower surface temperature on the tree reduces the severity of charcoalification and the degree of fire damage to the bark, thereby promoting the formation of large charcoal. The physiology of the tree only reduces the burning intensity on its surface, not that of the entire environment. And the combustion intensity of the surrounding fuel complex is not uniform and is constantly changing.

Lines 378-379:" This is not obvious from Figure 6, even if $p < 0.05$ is met. Distributions for all 4 Pinus species look fairly similar"

R: Thank you for the comment. We will discuss with next suggestion and the add to the revised manuscript.

Lines 380-383:" The lengths of particle images are highly unlikely to differ if both the widths and areas do not differ. Perhaps the significance threshold is just met for length (though not in all pairings) and just missed for width and area? Without the full statistical results we don't know. Some 'environmental variations' are offered as causes of this result, but no mechanisms are suggested to explain this."

R: Yes, we agree with you. This result can be attributed to a variety of factors. We will clarify and rephrase in the revised manuscript.

Lines 383-385:" The claim that all the samples come from crown fires is not made prior to this point. It obviously cannot apply to the grassland and herbaceous samples, and for the forest samples must surely mean surface fire + crown fire. The complete lack of information on the fires themselves is again a problem here, but I take this to mean that all the forest fires included crown fires. It does not follow from that that all the charcoal originates from bark."

R: Thank you for this comment. "All samples" refers to those collected from evergreen coniferous forests. And our point to emphasize that bark is one of the sources of charcoal, not the sole source.

Lines 387-388:" This is not possible. I think the intended claim is that the majority of the charcoal was bark charcoal, but this must be evidenced."

R: Thank you for this comment. while there is limited existing research on the primary sources of charcoal burning from wildfire, our field sampling and observations reveled that the primary source of charcoal is bark, particularly in vegetation types dominated by large plants.

Lines 390-392:" This seems like a non-sequitur. Is it the altitude or the canopy height that affects charcoal size, and how?"

R: Thanks. We will rephrase this sentence in the revised manuscript.

Lines 395-396:" It isn't clear what is meant by a higher degree of charcoalification, and neither of the references helps. I suspect the intended meaning is degree of fragmentation – i.e. a higher degree meaning smaller particles?"

R: Yes, thank you for your suggestion. We will correct this sentence.

Lines 400-401:" No explanation is given for how this occurs."

R: Thanks for the suggestion, we will add the about this in the revised manuscript.

Lines 404:" Complete combustion would mean no charcoal left to analyze"

R: Thanks, we will rephrase this.

Line 415-416:" The meaning of 'high charcoalification process' is unclear "

R: Thanks, we will clarify this.

Lines 419-421:" These are presumably the median or mean values, but there is no attempt to demonstrate that they are generally representative of these environments (e.g. though randomised sampling) or to estimate confidence intervals, so their interpretive value is limited"

R: Yes, they are mean values. our attempt to identify a threshold for distinguish different

vegetation types using this method proved. This result, in turn, underscores the limitations of the L/W.

Lines 422-424:" It would be more accurate to say 'contrasts with previous studies. Firstly, I do not think any cited author has claimed that fuel type can be determined from charcoal L/W ratio alone; the method is always used alongside others, and interpretations are environment-specific. If such a claim has been made, it should be cited here. Relatedly, the findings here are valid only for the environments studied (if at all), and cannot invalidate those of (e.g.) Umbanhowar & McGrath (1998) in the environments they studied. Finally, as described above, the method of measuring L/W by fitting rectangles will tend to underestimate variation, which may partly explain the lack of variation seen here."

R: Thanks, we will add some information. The method of measuring of fitting rectangles is be explained in the part of "Materials and Methods". We are not suggesting that the findings of other studies are invalid. Rather, we posit that the divergent results across studies are likely attributable to multiple factors, which warrants further investigation rather than outright dismissal of any particular outcome.

Lines 433-435:" This is already known, so it would be better to say "these findings provide further evidence...""

R: Thanks for your helpful suggestion. We will revise that.

Lines 439-447:" All this text should be in the Methods"

R: Thanks, we will change this to Methods.

Lines 450-452:" The claim being made here needs to be clarified. If the model allows charcoal measurements to be converted to vegetation types (at 72% accuracy), under what circumstances is it valid? How can the authors demonstrate that it is applicable beyond their own data set? Since only around 5% of particles are ultimately classified according to shape, the model relies mainly on particle size; therefore, the size fraction

used will be essential, but this has not been explained"

R: Thanks for your helpful suggestion. We will add the information to the revised manuscript.

Lines 455-457:" These statements remain unevidenced"

R: Thanks for the suggestion, we will modify this part.

Lines 480-484:" This was already understood, and yet it is the only generalized statement in the Conclusions (the rest being a restatement of the methods and results)"

R: Thanks for your helpful suggestion, we will rephrase the conclusion.

Figures 2-5:" There is no reference to the SEM images in the results or discussion, so they seem unnecessary. "

R: Thank you for pointing this. In our view, providing SEM images is crucial for summarizing the morphological characteristics of charcoal.

Figs 6-9:" Please explain exactly what is shown, as there are different conventions for box plots. How are the outliers defined? Does 'X' indicate the mean?"

R: Thank you for this remark. We will add the exactly explanation about the box diagram in the revised manuscript.

Fig 10:" The diagram needs to be explained in the caption. E.g., why are categories assigned to the root node and internal nodes?"

R: Thanks for your helpful suggestion, we will add this to the caption.

Technical Corrections:

Lines 175:" Please provide a reference for ImageJ, with details of the version used."

R: Thanks, we will provide the references.

Lines 213:" 'Charcoalification' should be a single word, without a hyphen"

R: Thanks, we will modify this.

Lines 335:" "Mustaphi and Pisaric,2014" should be "Courtney Mustaphi and Pisaric, 2014". (It

is given correctly in the reference list, though still under 'M'.)"

R: Thanks for your helpful suggestion, we will revise this.

Lines 355:" The citation "Crawford,2016" is not present in the reference list."

R: Thanks, we will add it in the reference list.

References list:

Crawford & Belcher (2014): The wrong journal is stated.

Zhang & Lv (2006): This should be Zhang & Lu. (The same error is made in the text.)

R: Thanks for your helpful suggestion, we will revise in the revised manuscript.